# Soil Water Erosion Modeling in Tunisia Using RUSLE and GIS Integrated Approaches and Geospatial Data

Mohamed Moncef Serbaji [1,*], Moncef Bouaziz [2] and Okba Weslati [3]

1 Laboratory for Environmental Engineering and Eco-Technology, National Engineering School of Sfax (ENIS), University of Sfax, Sfax 3038, Tunisia
2 Institute for Mine Surveying and Geodesy, Freiberg, University of Technology, 09599 Freiberg, Germany
3 Laboratory of Water, Energy & Environment, National Engineering School of Sfax (ENIS), University of Sfax, Sfax 3038, Tunisia
* Correspondence: mohamed-moncef.serbaji@enis.tn

**Abstract:** Soil erosion is an important environmental problem that can have various negative consequences, such as land degradation, which affects sustainable development and agricultural production, especially in developing countries like Tunisia. Moreover, soil erosion is a major problem around the world because of its effects on soil fertility by nutriment loss and siltation in water bodies. Apart from this, soil erosion by water is the most serious type of land loss in several regions both locally and globally. This study evaluated regional soil erosion risk through the derivation of appropriate factors, using the Revised Universal Soil Loss Equation (RUSLE), which was applied to establish a soil erosion risk map of the whole Tunisian territory and to identify the vulnerable areas of the country. The RUSLE model considers all the factors playing a major role in erosion processes, namely the erodibility of soils, topography, land use, rainfall erosivity, and anti-erosion farming practices. The equation is, thus, implemented under the Geographic Information System (GIS) "Arc GIS Desktop". The results indicated that Tunisia has a serious risk of soil water erosion, showing that 6.43% of the total area of the country is affected by a very high soil loss rate, estimated at more than 30 t/ha/year, and 4.20% is affected by high mean annual soil losses, ranging from 20 to 30 t/ha/year. The most eroded areas were identified in the southwestern, central, and western parts of the country. The spatial erosion map can be used as a decision support document to guide decision-makers towards better land management and provide the opportunity to develop management strategies for soil erosion prevention and control on the global scale of Tunisia.

**Keywords:** RUSLE model; GIS; soil water erosion; integrated approach; sustainable development; land degradation; vulnerable areas; soil loss rate

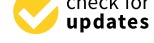



## 1. Introduction

Large-scale soil erosion is known to be one of the most severe problems that can lead to environmental damage [1,2]; consequently, affecting the political, social, and economic aspects of countries [3], particularly developing countries. The most dominant agent of soil erosion is water [4], which removes the soil surface materials through detachment, transportation, and deposition of the particles by rainfall and runoff. Soil erosion caused by water is defined as the breakdown of the soil structure and detachment of soil particles due to the falling raindrops and water flow exceeding a critical threshold [5]. The raindrops, by which the soil particles (sediments) are detached, hit the topsoil of the ground surface during rainfall. The detached sediments can be transported to rills. The rills gradually join together to form large channels that lead to gully erosion. The natural process of the removal of soil particles by water runoff, and their redistribution downslope, represents one of the main types of soil degradation [6]. It can cause negative consequences on soil productivity, human health, and the earth's environment, as well as on habitats, which can also be gradually degraded [7,8]. Many researchers have focused their studies on soil

erosion by water to assess soil erosion losses in different regions of the world using remote sensing and GIS technology [5,9–13]. Land Use and Land Cover (LULC) can influence the soil erosion process in a positive or negative way [14]. Human activities can accelerate the process of erosion, transport, and sedimentation through the urbanization, mining and construction of roads, highways, and dams [8].

More than 80 soil erosion models [15–23], with varying degrees of complexity, have been developed to evaluate potential soil loss for different spatial and temporal scales [24,25], such as the Universal Soil Loss Equation (USLE), which was developed by Wischmeier and Smith in 1965 to measure elementary experimental plots by respecting the exact dimensions [26]. However, this model had certain limitations since it only considered the sheet erosion processes at the scale of the plot [27–29]. To remove these limitations, the model has been improved, modified, and revised in several versions by integrating runoff factors to adapt it to the scale of a rainfall event [16]. Indeed, the USLE model has progressed to the Modified Universal Soil Loss Equation (MUSLE) model, which considers the topographic complexity through the use of the Digital Elevation Model (DEM) and anti-erosive practices. Finally, the Revised Universal Soil Loss Equation (RUSLE) improves the ability to determine the different factors involved in soil water erosion [30]. This new revised USLE maintains the basic structure of the USLE yet adds new algorithms to estimate and calculate the individual factors. The RUSLE model takes the form of a mathematical equation, which uses erosion factors as inputs to estimate the mean annual soil losses resulting from sheet and rill erosions [31]. It is an empirical model that groups the factors affecting the rate of water erosion, namely the kinetic energy of intense rainfall, soil properties, terrain characteristics, soil protection by vegetation cover, and anthropogenic practices. This model does not consider erosion processes such as detachment, transport, and deposition to estimate soil loss. The revised Wischmeier equation is combined with the GIS techniques to assess the rate of soil loss and the spatial distribution across different land covers. Thus, the advantages offered by GIS and geospatial data [11,32,33] can provide more efficient, more precise, and less time-consuming results [34], which helps to guide the decision-making. The RUSLE model is not used to estimate the amount of sediment migrating from a specific watershed, merely the amount of soil lost from any area [35]. For this, the model retains the same form as the equation used in the USLE model.

The RUSLE model involves the spatial combination of the different factors contributing to soil erosion. It was used to calculate soil losses (A), which is a multiplicative function that considers the erosivity and the aggressiveness of rainfall and runoff (R factor) (megajoules millimeter hour$^{-1}$ hectare$^{-1}$ year$^{-1}$), the soil erodibility (K factor) (ton hour megajoules$^{-1}$ mm$^{-1}$), and the resistance of the environment (C, P, and LS factors) (dimensionless). The LS in the RUSLE is generally the combination of L and S and represents the effect of the topography on erosion rates [36]. The RUSLE model has been used worldwide and adapted according to the climatic, topographic, and soil context [9,10,37,38]. It is still widely used to estimate soil erosion all over the world [39] and can be used to estimate soil erosion for large areas, including for entire countries [40].

Therefore, our study aimed to estimate the spatial distribution of soil erosion in Tunisia by modeling the soil water erosion using geospatial data and the integrated approaches of RUSLE and GIS, to contribute to a better understanding of the soil degradation in large-scale areas. The final spatial soil erosion map produced can serve as a useful input for deriving land planning and management strategies and provide an opportunity to develop a decision plan for soil erosion prevention, for better control and protection of both natural and man-made resources available in Tunisia.

## 2. Materials and Methods

### 2.1. Description of the Study Area

The Republic of Tunisia, located in North Africa, is situated between 30°13′ and 37°20′ north latitude and 7°32′ and 11°36′ east longitude (Figure 1). Furthermore, it is located in the southern part of the Mediterranean Sea and is bordered by Lybia in the southeast and

by Algeria in the west. The study area covers 155,084 km², including the two main islands (Jerba and Kerkennah), although without the other small islands, such as Kneiss, Zembra, Kuriat, Galite, and Chikly. Tunisia is the smallest country in North Africa, with a coastline in the north and east of around 1300 km.

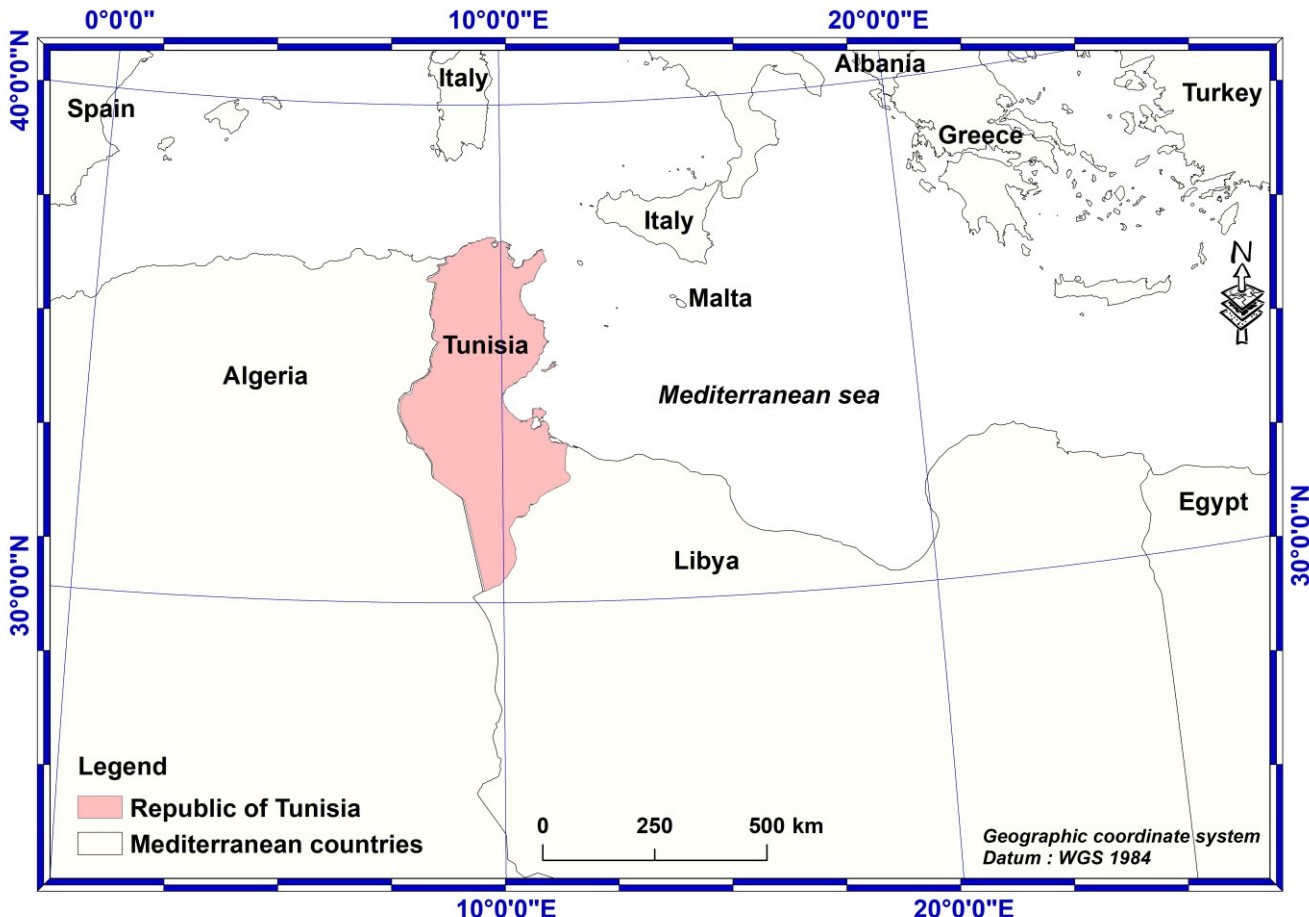

**Figure 1.** Map of the study area.

Tunisia has various types of landscapes, with mountainous areas in the northwest (Figure 2), upper and lower Steppe areas in the center, and wide plains in the east. The depressions of the great Chotts present the beginning of the Sahara in the south, with the mountains of Dhahar and the plains of Jeffara. The lowest altitude is 27 m below sea level, relating to the Chott areas, and the highest is 1549 m above sea level, in Jbel el Chaambi, while the whole study area has an average elevation of 256 m.

Tunisia has ten different bioclimatic zones (Figure 3), characterized by three factors: The mean annual precipitation, which decreases from north to south, the highest mean temperature of the hottest month (August), and the lowest mean temperature of the coldest month (January) [41]. The winter in Tunisia is soft and humid with temperatures between 8 °C and 15 °C, while the summer is hot and dry with temperatures between 22 °C and 35 °C, yet they can exceed 40 °C in August. Its northern part has a highly humid, low humid, and sub-humid bioclimate and an average annual rainfall, over the previous 31-years (from 1990 to 2020), of 650–736 mm. The central regions have a semi-arid higher, semi-arid middle, semi-arid lower, and arid-higher bioclimate with an annual rainfall of 250–650 mm. The southern part has an arid lower, Saharan higher, and Saharan lower bioclimate with an annual rainfall of 37–250 mm.

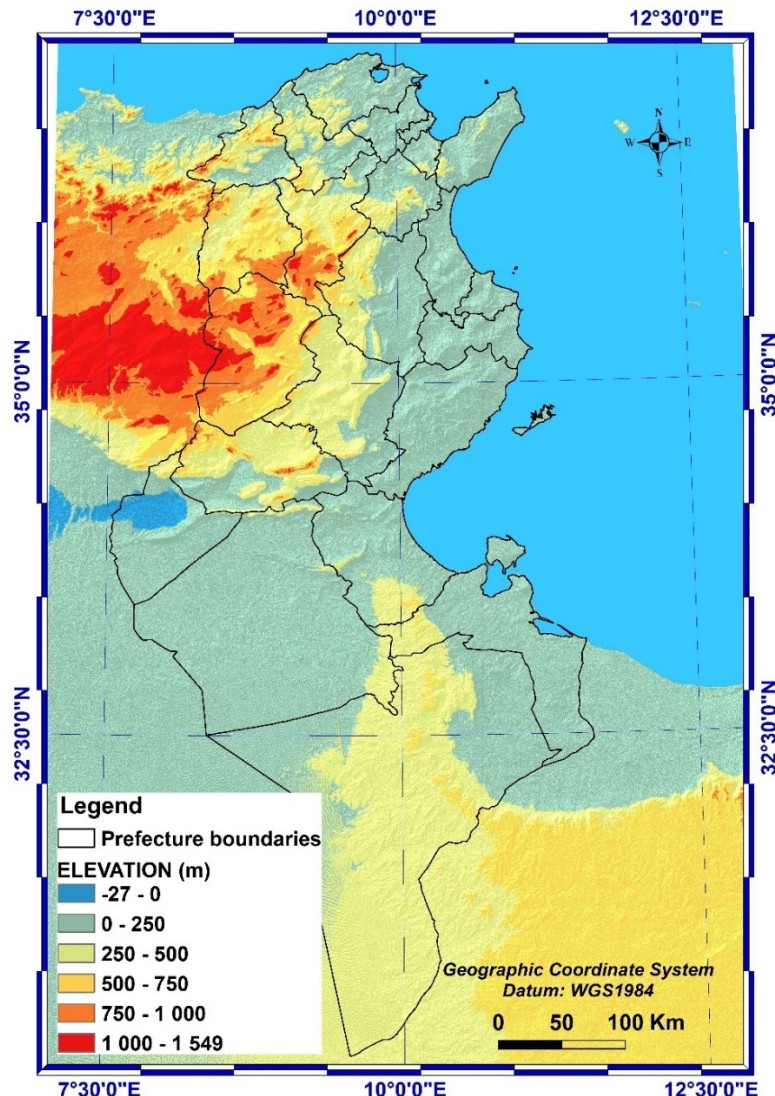

**Figure 2.** Distribution of altitudes in Tunisia generated from ASTER DEM data.

*2.2. Methodology and Data Source Processing*

The cartographic documents are generally produced on local scales. However, these data are often presented without recent updates, thus, are incomplete, and sometimes non-existent, such as geographical reference data on topography, precipitation, land cover, etc. Similarly, georeferenced digital data on a small scale are mostly scarce or obsolete. On the other hand, the global use of online databases has become available over these last few years. Furthermore, spatial analysis and modeling are facilitated mainly because of the existence of satellite data, which has become very easy to obtain continuously. Indeed, the raster mode, with a simple data structure and a regular shape of the grid, is very convenient to maintain geometric properties, which are common to all layers of information and, thus, facilitate the combination of different layers, since all numeric values are dependent on the same base unit, the pixel.

In addition, these data need to be tested in several case studies on a global scale. Hence, one of the objectives of this study was to assess the potential of the existing digital data for spatialized modeling in a GIS of soil water erosion in the current Tunisian context. It can be, in some cases, considered as an alternative, with which to overcome the problem of a lack of data.

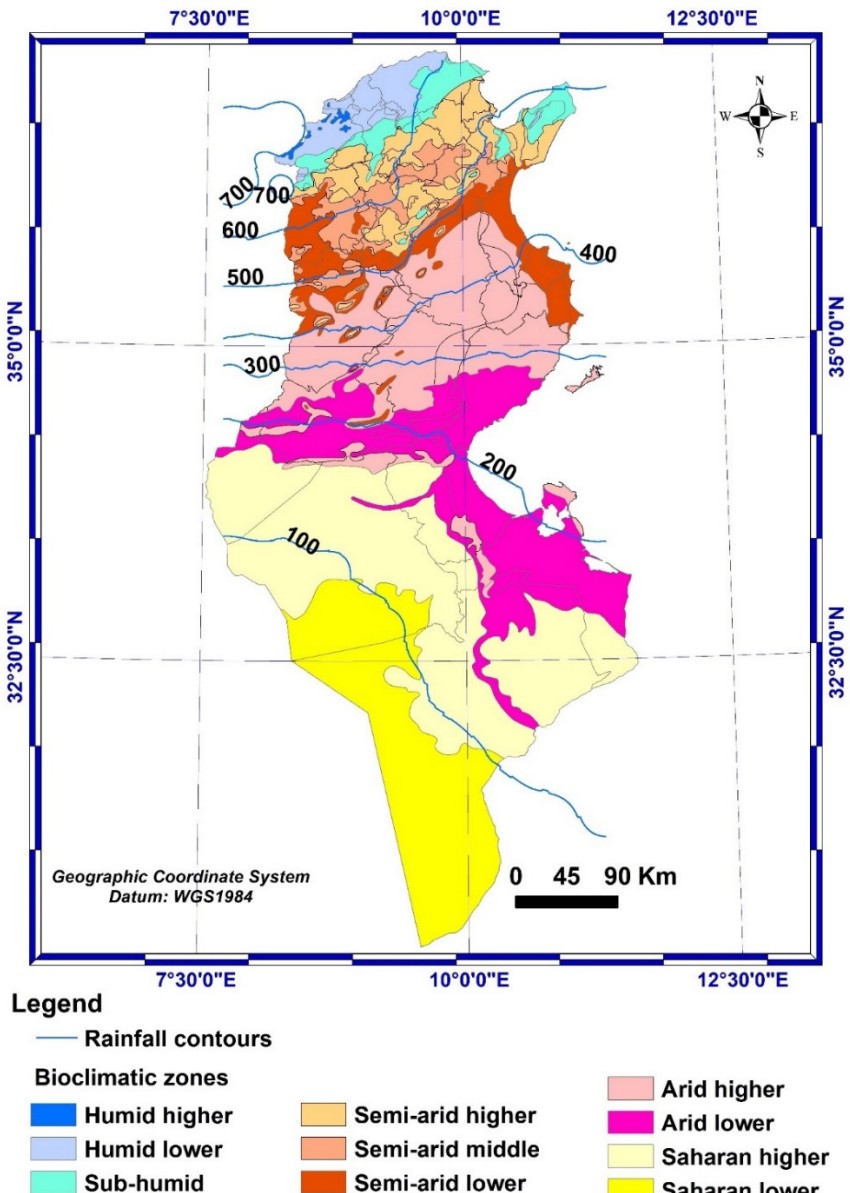

**Figure 3.** Bioclimatic map of Tunisia.

The choice to select the RUSLE model (Equation (1)) in this study was motivated by the fact that this model does not differ conceptually from the original soil loss equation (USLE) of Wischmeier and Smith [26]. On the contrary, it improves the quality of the environmental parameters that define the role of each factor. Moreover, the input data for this model is easier to access compared to other more recent models, which require more sophisticated data.

The RUSLE is written as:

$$A = R * K * LS * C * P \tag{1}$$

where A is the soil loss per unit area, expressed in the units of K and the period selected for R;

R is the rainfall and runoff erosivity factor;

K is the soil erodibility factor;

LS is the slope length and slope steepness factor;

C is the crop management factor;

P is the support practice factor.

The RUSLE model was used to estimate the yearly soil loss average from water erosion, by considering the five factors indicated in Equation (1). The latter has been applied using the GIS and geospatial data and is composed of: (1) Climatic data including a 31-year average annual precipitation, downloaded from the NASA-POWER meteorological parameters, taken from NASA's MERRA-2 assimilation model, and which is necessary for computing the rainfall and Runoff Erosivity factor (R). (2) The soil map of Tunisia extracted from the FAO Digital Soil Map of the World (DSMW), essential for calculating the soil erodibility factor (K). (3) The Global Digital Elevation Model (ASTER GDEM), which is used to calculate the topographic factor (LS). (4) The LULC map of Tunisia extracted from ESRI 2020 Global Land Cover and used to calculate and map the vegetative cover factor (C), as well as the support and management practice factor (P).

Each of the factors was derived separately in a raster format based on rainfall pattern, soil type, topography, and LULC data, in the context of soil water erosion modeling and in order to generate global scale maps. However, the current study, in deal with the estimation of soil erosion by water, did not consider the development stage of the crops. Furthermore, the season variability effects were not considered.

The geospatial data with the various types and origins are listed in Table A1 (Appendix A). Figure 4 shows the flowchart of the method used to estimate each factor and for the analysis of soil loss based on the RUSLE model and GIS technique.

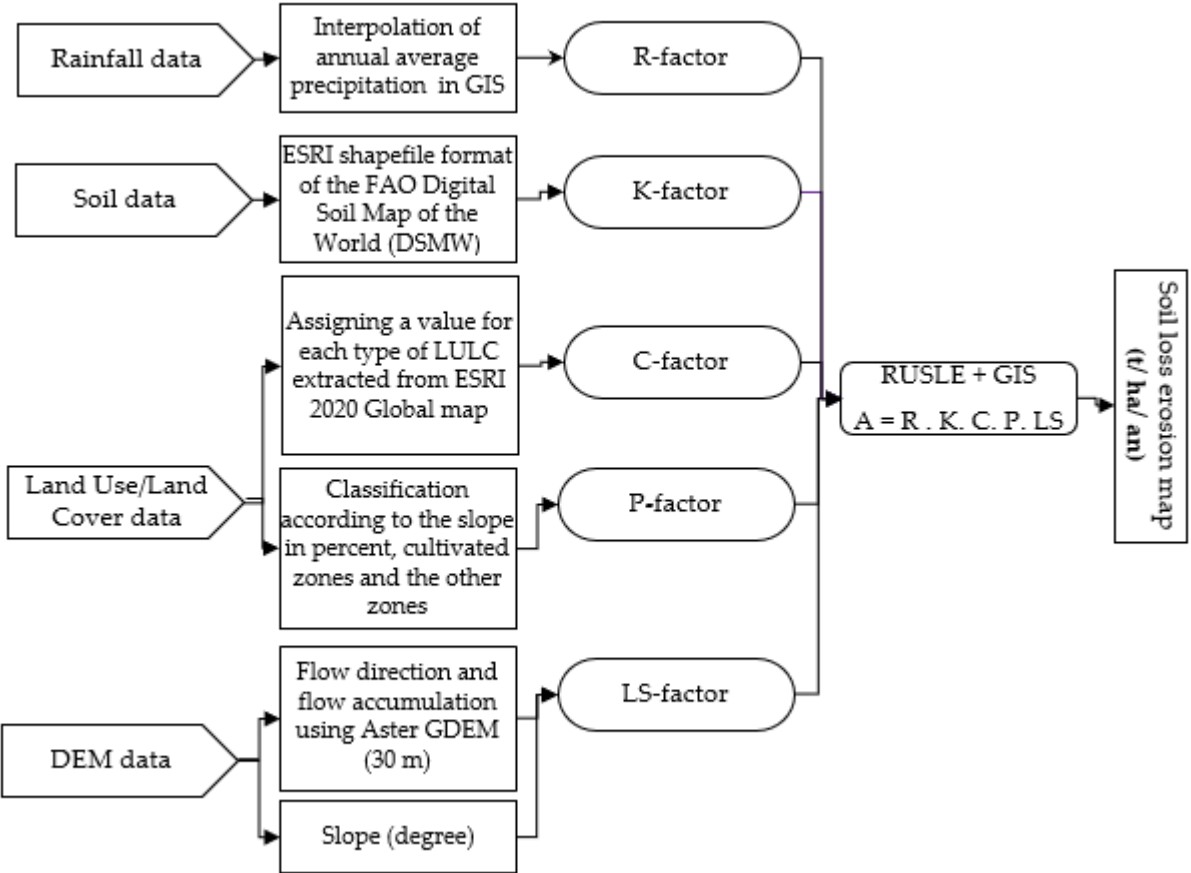

**Figure 4.** Flow chart for the analysis of soil loss based on the RUSLE model and GIS technique.

The following parts of this article provide a brief description of the various factors shown in Equation (1) and present the results obtained after modeling and analyzing the geospatial data.

### 2.2.1. Rainfall and Runoff Erosivity Factor (R-Factor)

The estimation of the R-factor requires knowledge of the kinetic energies and the average intensity over 30 min of the raindrops that fall each time, over a period of up to 30 years [42]. However, this direct method requires precipitation records at high resolutions to calculate the R-factor. Other authors have developed alternative formulas when these data are not available. The equations frequently used to calculate the R-factor, which only use the annual precipitation are those of Renard and Freimund [43] (Equations (2) and (3)), whose expressions are:

$$\text{R} = 0.0483 \cdot P_i^{1.61} \qquad\qquad \text{if } P_i < 850 \text{ mm} \qquad\qquad (2)$$

$$\text{R} = 587.8 - 1.219 \, P_i + 0.004105 \cdot P_i^2 \qquad\qquad \text{if } P_i > 850 \text{ mm} \qquad\qquad (3)$$

where R is the measure of rainfall erosivity and $P_i$ is the average annual precipitation (mm).

The estimations of rainfall and runoff erosivity using rainfall data with long-time intervals have been conducted by many authors for different regions of the world [34,44]. To calculate the average annual R-factor values in this study, 31-year annual data averages have been used and an interpolation of this data was applied to produce a representative rainfall distribution map, which is used as an input to calculate the R-factor, knowing that the rainfall data available for the study area is not homogenous. This parameter, expressed in (MJ·mm·ha$^{-1}$·h$^{-1}$·year$^{-1}$), takes into consideration the influence of the climatic aggressiveness on soil loss [45].

To generate the R-factor for the whole of Tunisia, a 31-year average annual precipitation from 1990 to 2020 was downloaded freely, as a comma-separated values (CSV) file, from the National Aeronautics and Space Administration (NASA) Prediction of Worldwide Energy Resources project, known as the POWER project, which was initiated in 2003. It provides access to solar and meteorological datasets for the entire world, particularly over the regions where the surface measurements are sparse or nonexistent. The meteorological parameters are based on the MERRA-2 assimilation model. The monthly and annual precipitation data are derived from NASA's Global Precipitation Measurement (GPM) and interpolation of this data was applied to provide a representative rainfall distribution map, which was used as an input to calculate the R-factor. Since the mean annual precipitation in Tunisia does not exceed 850 mm, equation 3 was used to calculate the rainfall–runoff erosivity factor, considering the bioclimatic map of Tunisia, which has been developed for climatically homogeneous areas. The inverse distance interpolation (IDW) method was used to minimize the calculation error and, above all, to optimize the processing time on the ArcGIS software. It is a method that consists of creating, from the aggregated points and the corresponding information, a value grid with a certain continuity.

### 2.2.2. Soil Erodibility (K-Factor)

The soil erodibility, K, expressed in [t·h·MJ$^{-1}$·mm$^{-1}$], determines the resistance of different types of soil to erosion, knowing that soils are more or less sensitive to water erosion. The K-factor is determined according to the characteristics of the soil: infiltration capacity, retention texture, and susceptibility to particle removal. The infiltration and the high cohesion of the materials increase the soil's resistance to the removal and gullying of particles. The high rate of sand stabilizes the structure of the soil and makes it less sensitive to climatic aggression. Similarly, organic matter improves the physical and chemical properties (cohesion, structural stability, and porosity) of the soil, increasing the ability to retain water, and strengthening its resistance to erosion [46]. Thus, the higher the percentage of sand, the more the soil is permeable, which implies a low valued K-factor and vice versa. Bolline and Rousseau (1978) [47] established the classification, while Table 1 presents the interpreted soil susceptibility indexes.

However, soil erodibility is not constant and varies with soil type, seasons, and cultivation techniques. The soil map of Tunisia was prepared using the FAO Digital Soil Map of the World Shapefile (DSMW), where the soil data layer has been clipped, according to the

boundaries of the country relative to the study area, in the ArcGIS Desktop environment. The K-factor was calculated using the Williams (1995) [48] formula (Equation (4)) and FAO Digital Soil Map.

**Table 1.** Soil susceptibility index classification (Bolline and Rousseau, 1978) [47].

| Erodibility (K) | Type of Soil |
|---|---|
| K < 0.10 | Soil highly resistant to erosion |
| 0.10–0.25 | Soil fairly resistant to erosion |
| 0.25–0.35 | Soil moderately resistant to erosion |
| 0.35–0.45 | Soil with low erosion resistance |
| >0.45 | Soil with very low resistance to erosion |

The raster dataset of the FAO/UNESCO Digital Soil Map of the World (DSMW) has a spatial resolution of $5 * 5$ arc minutes and is in geographic projection.

$$K = f_{csand} * f_{cl-si} * f_{orgC} * f_{hisand} \tag{4}$$

where:

$f_{csand}$ is a factor that produces low soil erodibility factors for soils with high coarse-sand content and high values for soil with little sand.

$f_{cl-si}$ is a factor that presents low soil erodibility factors for soils with high clay to silt ratios.

$f_{orgC}$ is a factor that reduces soil erodibility for soils with extremely high sand content.

$f_{hisand}$ is a factor that reduces the soil erodibility factor with highly coarse sand contents and high values for soils with little sand.

The factors are calculated using these equations (Equations (5)–(8))

$$f_{csand} = 0.2 + 0.3 \exp\left[-0.256 m_s \left(1 - \frac{m_{silt}}{100}\right)\right] \tag{5}$$

$$f_{cl} - f_{si} = \left(\frac{m_{silt}}{m_c - m_{silt}}\right)^{0.3} \tag{6}$$

$$f_{orgC} = 1 - \frac{0.25 * orgC}{orgC + \exp[3.72 - 2.95 orgC]} \tag{7}$$

$$f_{hisand} = 1 - \frac{0.7\left(1 - \frac{m_s}{100}\right)}{1 - \frac{m_s}{100} + \exp\left[-5.57 + 22.9\left(1 - \frac{m_s}{100}\right)\right]} \tag{8}$$

where $m_s$ is the percent sand content (0.05–2.0) mm diameter particles,

$m_{silt}$ is the percent silt content (<0.002) mm diameter particles,

$m_c$ is the percent clay content (0.05–2.0) mm diameter particles,

orgC is the percent organic carbon of the layer (%).

2.2.3. Topographic Factor (LS-Factor)

The slope gradient and slope length factor are the two parameters that constitute the topographic factor, these factors were estimated, in this study, through an ASTER Global Digital Elevation Model (ASTER GDEM) with a 30 m resolution, which was downloaded freely from (https://search.earthdata.nasa.gov/downloads/) (Accessed on 12 February 2022) [49]. The DEM of Tunisia's territory is ranged from −27 m to 1549 m (Figure 2). The latter value was incorporated into the GIS to determine accurately the slope gradient (S) and slope length (L); the effect of slope length and degree of slope interaction should always be considered together [50]. Thus, to create the slope length and steepness (LS-factor), equivalent to the topographic factor and relief factor, respectively, firstly the filling process was performed on the DEM data using the GIS extension Arc Hydro Tools, then, the flow direction and flow accumulation maps were created. The flow accumulation was determined for each cell using the flow that passes through that cell. The greater the flow

accumulation value, the easier the area forms the runoff and vice versa. The slope map, in degrees, required to estimate the LS factor, shows that most of the slopes in Tunisia ranged from 0 to 10°, which represents about 95.54 % of its total area (Figure 5).

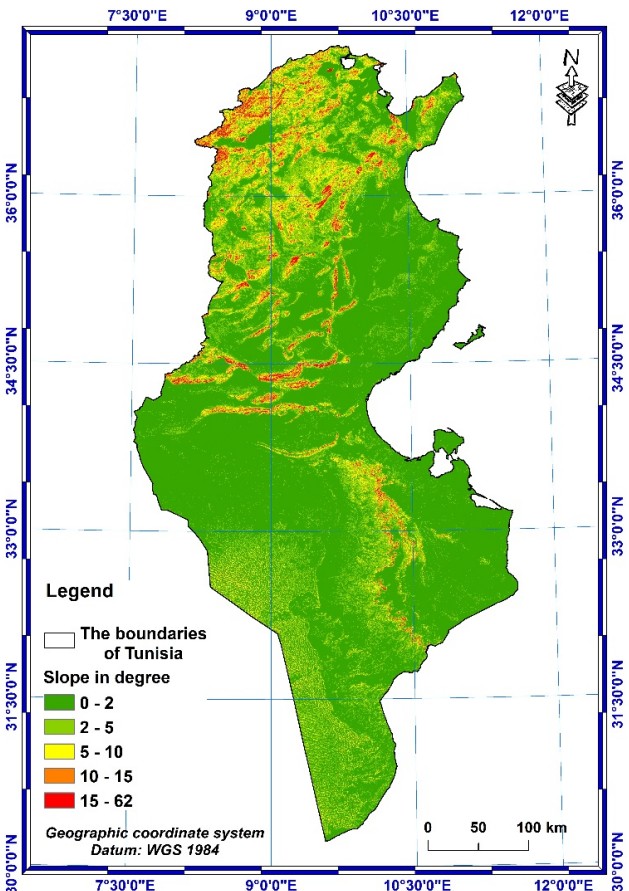

**Figure 5.** Slope map of Tunisia.

In 1985, Moore and Burch [51] developed the equation below (Equation (9)) to compute the length–slope factor:

$$LS = \left( \frac{As}{22.13} \right)^m \times \left( \frac{\sin \theta}{0.0896} \right)^n \tag{9}$$

Equation (9) has previously been applied by many researchers [50,52,53]. The exponent value, m, can be taken as equal to 0.4, while the value of 'n' can be taken to equal 1.3 [52–54].

Where:

*LS*: is slope steepness–length factor.

As: is the flow accumulation (in meters).

θ: is the slope angle (in radians).

m = 0.4–0.6 and *n* = 1.2–1.3.

In general, as the length of the slope increases, the total soil erosion and the soil erosion per unit area increase due to the gradual accumulation of runoff water in the downhill direction of the slope. As the steepness of the slope increases, the velocity and erosivity of the runoff increase.

### 2.2.4. Vegetative Cover Factor (C-Factor)

It is a dimensionless factor presenting the effectiveness of vegetation cover in relation to the susceptibility of the soil to erosion [55]. Vegetation cover and its spatial distribution play an important role in reducing the effects of the runoff by amortizing the impact of rainwater on an area [15].

According to several studies, the RUSLE factor, which ranges from zero for water and very well-protected soil with very strong coverage effects, to one, for a surface that produces a lot of runoffs (bare soil) and where the soil is very susceptible to water erosion [56]. Several authors consider that the C-factor to be around 0.01 (1/100) under dense forests, 0.05 (5/100) under grasslands, and 0.24 (24/100) under crops (Table 2). Other researchers have adopted a new simplified approach for calculation of the C-factor, which uses remote sensing techniques, such as the classification of satellite images [57,58] and vegetation indices [59], or the use of the Land Use–Land Cover (LULC) map and class assignments for each entity [60]. In fact, the C-factor reflects the effect of LULC, cropping, and management practices on the rate of soil erosion [61–64].

**Table 2.** C-factor value of different soil types.

| LULC Type | C-Factor | Source |
|---|---|---|
| Cropland | 0.24 | Guo et al., 2015 [65] |
| Forest (Dense) | 0.01 | Hurni, 1985 [66] |
| Grassland | 0.05 | Tiruneh and Ayalew, 2015 [67] |
| Shrubland | 0.2 | Tiruneh and Ayalew, 2015 [67] |
| Bare land | 0.6 | Ewunetu et al., 2021 [68] |
| Waterbody | 0 | Erdogan et al., 2006 [69]; Swarnkar el al., 2018 [70] |
| Settlement | 0.15 | Hurni, 1985 [66] |

The C-factor in this study was determined based on the literature, by assigning a value for each type of LULC extracted from the ESRI 2020 Global map of LULC, derived from ESA Sentinel-2 imagery at 10 m resolution, and downloaded freely from https://livingatlas.arcgis.com/landcover/ (accessed on 13 February 2023) for all land masses on the planet. Two Individual GeoTIFF scenes (32S_20200101-20210101 and 32R_20200101-20210101) covering the Tunisian territory, were downloaded from the Esri 2020 Land Cover Downloader application. These scenes were joined to form a mosaic image and the values of the C-factor (Table 2) for 7 class LULC types, covering the study area, were used in the data analysis to build the C-factor map for the year 2020 at 10 m resolution. The analysis was performed in ArcGIS and then converted to a grid with a 30 × 30 m spatial resolution.

### 2.2.5. Support and Management Practice Factor (P-Factor)

This factor, which is dimensionless, represents the soil protection based on anti-erosion cultivation techniques that reduce the runoff speed and, thus, reduce the risk of water erosion. It varies according to the landscaping carried out, namely cultivation on a level curve, in alternating strips or on terraces, reforestation in benches, and ridging [15].

The P-factor is between 0 and 1, in which the value 0 represents a very good environment of resistance to erosion and the value 1 shows an absence of anti-erosion practices [71]. Wischmeier and Smith (1978) [15] established the classification according to the slope in percent (Table 3), showing that this factor can be distributed according to two zones: the cultivated zones and the other zones.

**Table 3.** P-factor reference values adopted from Wischmeier and Smith (1978) [15].

| | Slope (%) | P-Factor |
|---|---|---|
| | 0–5 | 0.1 |
| | 5–10 | 0.12 |
| Agricultural land | 10–20 | 0.14 |
| | 20–30 | 0.19 |
| | 30–50 | 0.25 |
| | 50–100 | 0.33 |
| Other land | All | 1 |



## 3. Results and Discussion

In this part, the results of the processing and the calculations of each factor made for the spatialization of the soil loss rates in the study area will be presented in more detail. After applying the RUSLE model, an assessment of the impacts of erosion at the regional scale in Tunisia will also be detailed in this section.

### 3.1. Spatial Distributions of RUSLE Factors

#### 3.1.1. R-Factor

The rainfall erosivity factor quantifies the effects of rainfall aggressiveness and, therefore, reflects the amount and rate of runoff associated with a rainfall event. It was calculated from the annual average precipitation data recorded over a time interval of 31-years (from 1990 to 2020).

After its calculation, it appears that the values of the R-factor in Tunisia range from 24.29 to 1859.84 MJ·mm·h$^{-1}$·ha$^{-1}$·year$^{-1}$. However, the average value in the entire country is 473.73 MJ·mm·h$^{-1}$·ha$^{-1}$·year$^{-1}$. The maximum value is observed in the northwest region of Tunisia and the minimum value is observed in the south.

The maps in Figures 6 and 7 show that the precipitation and erosivity factor (R-factor) decrease gradually from the south towards the extreme northwestern part of Tunisia. In addition, it is important to note that rainfall and the runoff erosivity factor are crucial factors in assessing soil erosion for future LULC and climate changes.

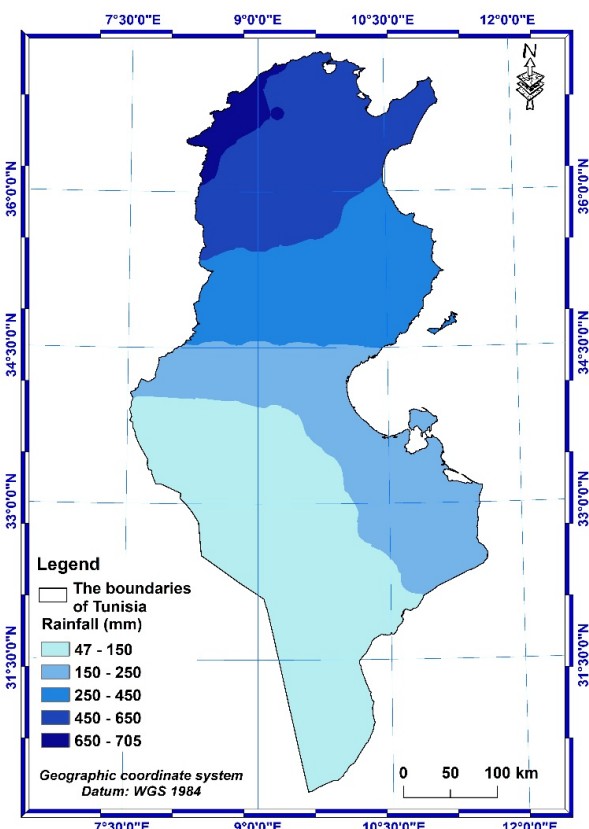

**Figure 6.** Map of mean annual rainfall in Tunisia.

#### 3.1.2. K-Factor

The physical–chemical properties of soils, such as organic matter, fine sand, clay, and silt contents were obtained from the FAO-UNESCO Soil Map of the World. The percentage of organic matter (OM) depended on the presence and nature of the vegetation on the ground, as well as how the vegetation was used.

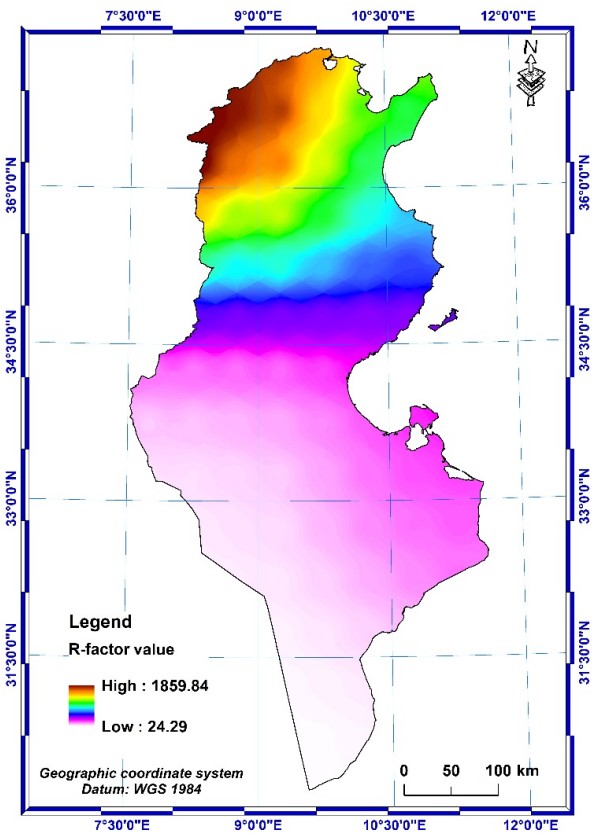

**Figure 7.** R-factor spatial distribution in Tunisia.

Determining the K-factor was performed using Equations (4)–(8) from Williams (1995) [48] and the soil map of Tunisia (Figure 8), extracted from the FAO-UNESCO Digital Soil Map of the World (DSMW) shapefile [72]. The soil erodibility spatial distribution map (Figure 9) shows that the values of the K-factor range from 0.10 to 0.19 t·h·MJ$^{-1}$·mm$^{-1}$. According to the classification of Bolline and Rosseau (1978) [47], the soils in Tunisia are, therefore, all classified as "soil fairly resistant to erosion".

### 3.1.3. LS-Factor

The slope map (Figure 5) was obtained by the Slope tool in the Spatial analyst and was then used to calculate the steepness of the slopes. The mean value of the slopes throughout the country is 2.24°, with a standard deviation of 3.50°, while values above 10° are possessed, especially in the southeastern, central, and northwestern areas of the country.

Knowing that the LS factor is the derivative product of Equation (9) (Moore and Burch, 1985) [51], the equation of this product was formulated on Map Algebra's Raster Calculator following the syntax:

LS = Power ("Flow Accumulation" * Cell size/22.13,0.4) * Power (Sin ("Slope in degree" * 0.0174533)/0.0896,1.3)

Hence, "Slope in degree" represents the slope map, as represented by Figure 5, and the cell size represents the spatial resolution of the ASTER GDEM and is equal to 30 m. The slope was converted to radians by multiplying by 0.0174533, since 1 degree = 0.0174533 radians.

Thus, the following LS topographic factor map was obtained (Figure 10), showing that the value range of the LS factor is between 0 and 12.41. However, the average value in the whole country is 1.09.

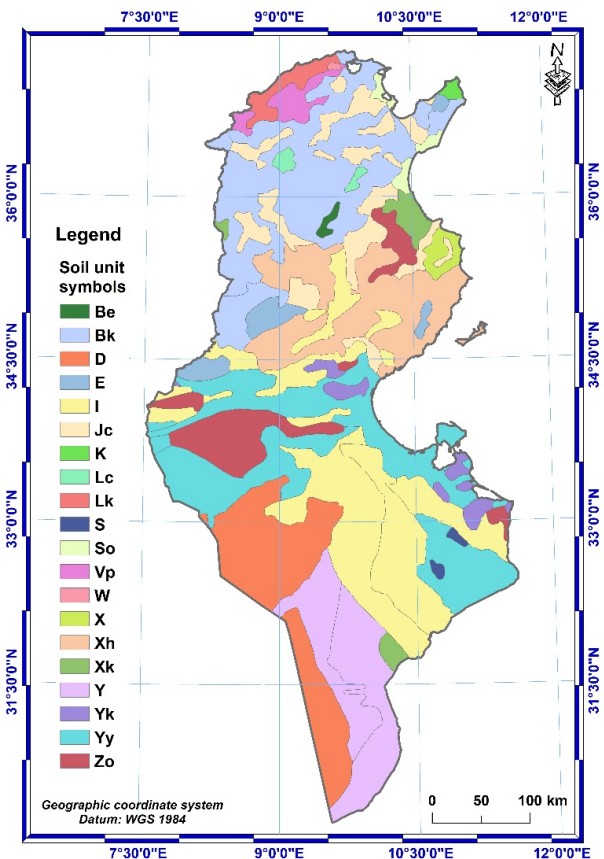

**Figure 8.** Soil map of Tunisia (extracted from the FAO-UNESCO Soil Map of the World).

### 3.1.4. C-Factor

The vegetation cover factor map shows that the values vary from 0 to 1 over the whole country. The minimum values are found in the northern part of the country, where according to the LULC map (Figure 11), there are mainly dense forests and cultivated areas corresponding to well-protected soil with very strong coverage effects. On the other hand, as we progress south, the value of the C-factor (Figure 12) is close to 1, especially on bare soil that produces a lot of runoffs and leaves the soil very susceptible to water erosion.

### 3.1.5. P-Factor

The support and management practice factor explains the effects of agricultural practices that can minimize the impact of rainwater and reduce the rate of runoff, thereby logically reducing soil loss (Wischmeier and Smith, 1978) [15].

The determination of the P-factor required a map of the agricultural and nonagricultural land to be established (Figure 13) from the LULC map of Tunisia, previously extracted from the ESRI 2020 Global map, and derived from ESA Sentinel-2, as well as the slope map in percent.

The values of the P-factor in agricultural areas decreased as the slope decreased and vice versa on steep terrain. These values vary from 0.1 on low-sloped land to 0.33 on very steep land. The other nonagricultural areas are all classified as areas without erosion control practices, which are assigned the value P = 1 (Figure 14).

The results of the calculations for each factor can be summarized as follows:

- The values of the erosivity factor on the rains vary from 24.29 to 1859.84 MJ·mm·h$^{-1}$· ha$^{-1}$·year$^{-1}$, with an average of 473.73 MJ·mm·h$^{-1}$·ha$^{-1}$·year$^{-1}$ over the whole country.
- The soil erodibility K-factor is classified as "fairly resistant to erosion", with values varying from 0.10 to 0.19 t·h·MJ$^{-1}$·mm$^{-1}$.

- The value range for the topographic factor, LS, is between 0 and 12.41, with an average value of 1.09 over the whole country.
- The values of the vegetative cover C-factor vary from 0 to 1.
- The values of the support and management practice factor vary from 0.1 to 0.33 for agricultural land and are equal to 1 for nonagricultural land.

### 3.2. Calculation of Soil Loss Rate and Quantification of Erosion

The modeling of the factors involved in the water erosion of soils has enabled the quantification and spatialization of this phenomenon.

The RUSLE model gives an approximation of the average value of soil loss, expressed in tons per hectare per year, on a given scale of study. This approximation is based on the joint product of the five factors described above. In this study, the modeling is essentially carried out on ArcGIS. To estimate and map the spatial distribution of soil losses for the entire Tunisian territory, the maps of the five factors, seen in the previous sections, were projected according to the same coordinate system "Universal Transverse Mercator (UTM) zone 32 N, using the Datum: WGS 1984", with a spatial resolution of 30 m × 30 m for each. Then, we proceeded to calculate pixel by pixel using the RUSLE (1) with the "Raster calculator" tool, and the "Map Algebra" function in ArcMap.

The application of this model allowed the development of the map displaying the average values and the distribution of soil loss rates for the whole country (Figure 15).

By carrying out the calculation according to the empirical and spatialized model of the RUSLE on ArcGIS, the average value range of soil losses in Tunisia was obtained, which varied from 0 to a maximum value of 619.44 t/ha/year. These results have been subdivided into five classes, according to Kefi et al. (2012) [9], and are represented in Table 4.

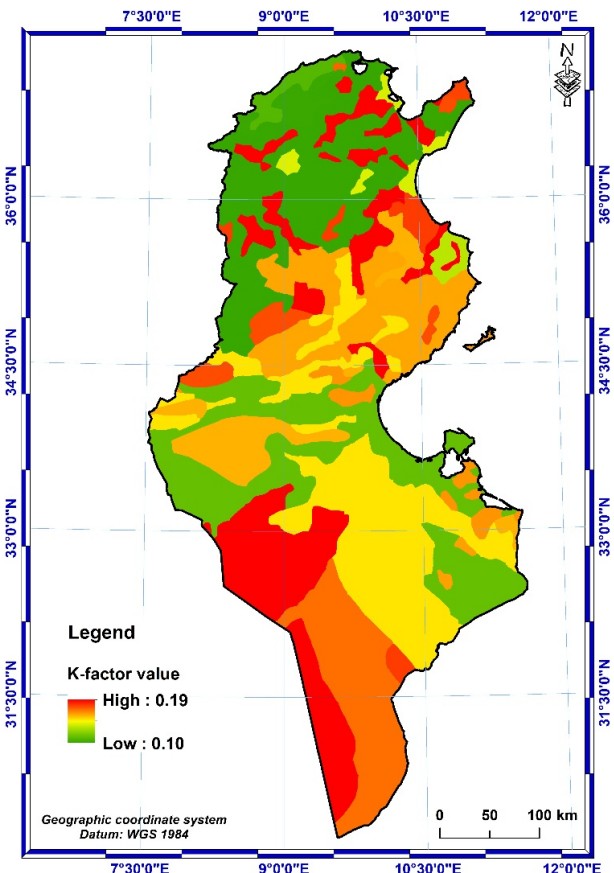

**Figure 9.** K-factor spatial distribution.

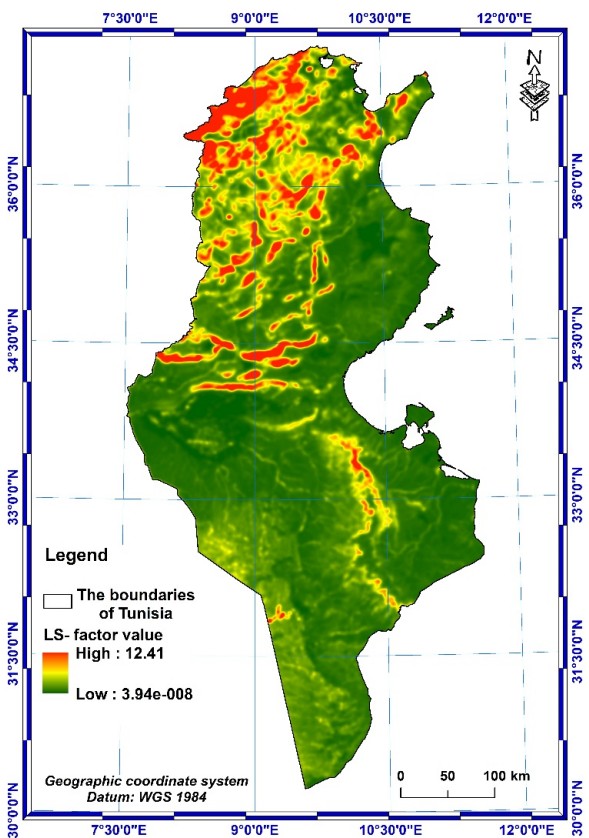

**Figure 10.** LS-factor spatial distribution.

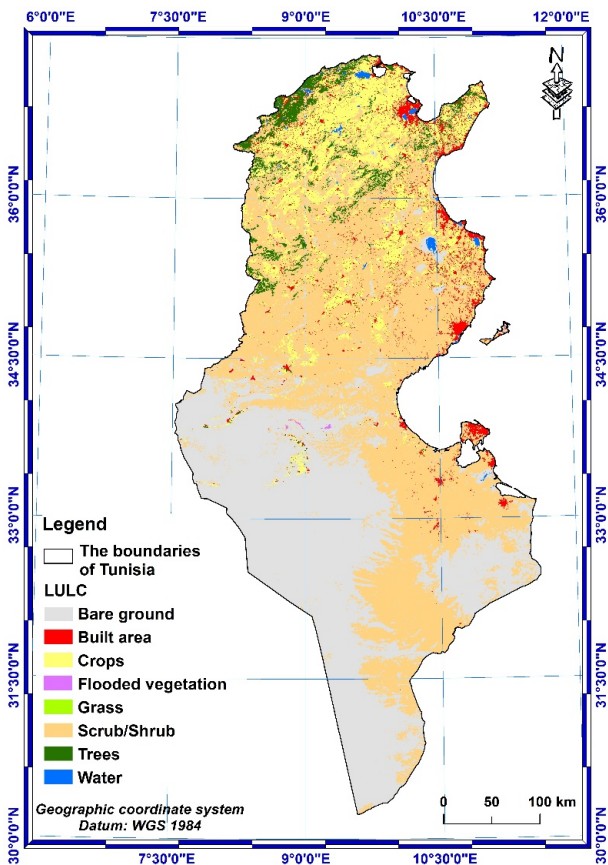

**Figure 11.** Land Use–Land Cover map of Tunisia.

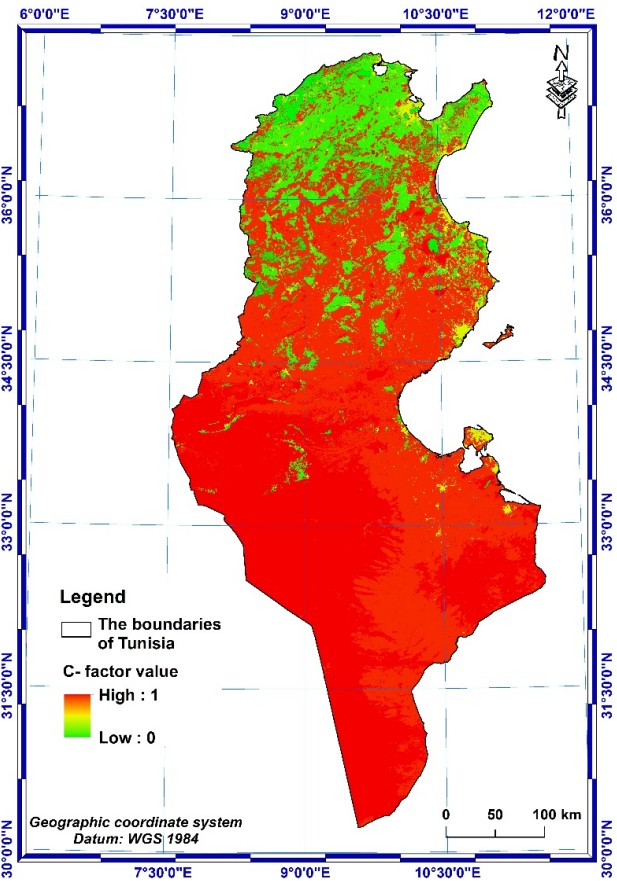

**Figure 12.** C-factor spatial distribution.

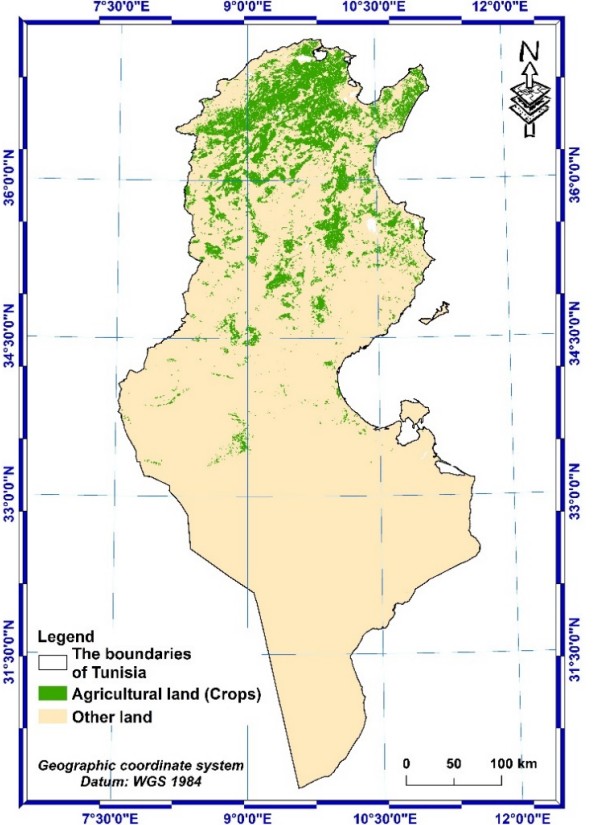

**Figure 13.** Agricultural and nonagricultural lands in Tunisia.

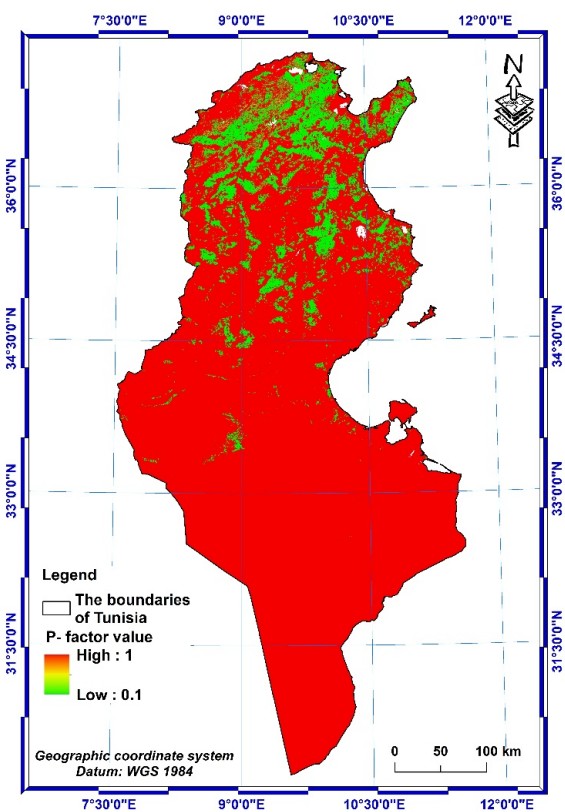

**Figure 14.** P-factor spatial distribution.

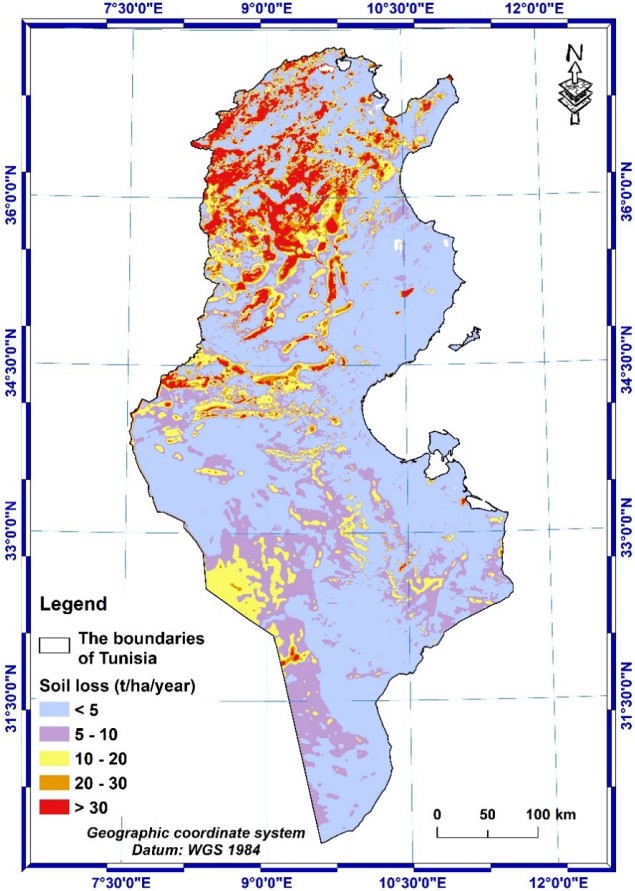

**Figure 15.** Soil loss map in Tunisia.

**Table 4.** Soil loss distribution.

| Score Scale | Soil Loss Class (t/ha/year) | Area (km$^2$) | Area Percentage (%) | Indicator |
|:---:|:---:|:---:|:---:|:---:|
| 1 | <5 | 87,559.33 | 56.34 | Very low |
| 2 | 5–10 | 32,854.00 | 21.14 | Low |
| 3 | 10–20 | 18,471.60 | 11.89 | Moderate |
| 4 | 20–30 | 6535.03 | 4.21 | High |
| 5 | >30 | 9989.66 | 6.43 | Very high |

Despite a large domination of very low soil erosion by water, representing 56.34% of the total area of the Tunisian territory, 10.64% of the total area of the soil is severely exposed to water erosion (high and very high soil erosion).

## 4. Conclusions

This study was the first to provide a complete and detailed water soil erosion map for Tunisia on a whole-country scale. The RUSLE model, an empirical model used to assess soil losses per year, has been conducted to evaluate the soil losses in Tunisia, which were mapped at a 30 m cell size. The usage of the RUSLE was significantly facilitated by the deployment in the Geographic Information Systems (GIS) and the use of geospatial data, provided over vast regions, such as the global territory of Tunisia. Therefore, the use of this approach contributed to regional soil erosion risk assessments through the rapid derivation of appropriate indices.

Therefore, this study identified an adequate method that integrates geospatial data and GIS with the RUSLE model to estimate the spatial distribution of soil water erosion in the whole of Tunisia and provided a spatial erosion map. The quantification of water soil erosion losses is an essential approach to spatialize the zones most sensitive to water erosion. Consequently, this approach can provide, in practice, relevant results for the potential evaluation of soil losses at the regional scale and make an important contribution to regional soil erosion risk assessments through the derivation of appropriate factors, despite the few studies that have been carried out for the modeling of soil erosion at the scale of countries around the world.

Soil erosion by water is a natural phenomenon that may or may not be accelerated by various agents. The impact of soil erosion depends on several parameters. In this study, the rates of soil loss by water erosion were listed according to five different classes, including, very low, low, moderate, high, and very high.

There is normally a tolerable amount of soil loss in the natural cycle of the Earth's elements, meaning that soils can maintain their long-term productivity. Tolerable soil loss is the maximum annual amount of soil that can be removed without affecting the long-term natural productivity of soil covering the slope of an area. The limit value is set in this study at 10 t/ha/year.

The results indicated that 6.43% of the surface of Tunisia, which corresponds to 9989.66 km$^2$, was affected by a very high soil loss rate (>30 t ha$^{-1}$ y$^{-1}$). Moreover, 4.20% of the total area of the country, which corresponds to 6535.02 km$^2$, was affected by high soil loss rates, ranging from 20 to 30 (t ha$^{-1}$ y$^{-1}$). These results can be useful in identifying the vulnerable areas of the country and developing a less time-consuming decision plan, which is more efficient for soil erosion prevention and control. This integrated approach, based on the GIS and erosion models, can provide useful contributions and a better understanding of the soil degradation of large-scale areas in Tunisia, to better preserve and protect both the natural and man-made resources that are available in the country. The quantitative evaluation of the spatial distribution of the erosional area and erosion factor analyses are important for future decision-making [73]. According to the study results, great efforts are needed to establish suitable strategies and measures to preserve water and soil resources and enhance ecological restoration in Tunisia. For future research, it is necessary to estimate additional

kinds of soil erosions, such as wind erosion, which is an important factor in the southern part of Tunisia.

**Author Contributions:** Conceptualization, M.M.S. and M.B.; methodology, O.W.; formal analysis, M.M.S., M.B. and O.W.; writing—original draft preparation, M.M.S.; writing—review and editing, M.M.S., M.B. and O.W. All authors have read and agreed to the published version of the manuscript.

**Funding:** This research received no external funding.

**Data Availability Statement:** The research data is presented mainly in Table A1. The other dada can be provided by contacting the authors directly.

**Conflicts of Interest:** The authors declare no conflict of interest.

## Appendix A

**Table A1.** Description of the data sources.

| N° | Type of Data | Data Description | Name of the Service That Provide the Data | Link |
|---|---|---|---|---|
| 1 | Rainfall data | Monthly and annual precipitation data derived from NASA's Global Precipitation Measurement (GPM)-CSV file format | National Aeronautics and Space Administration (NASA) Prediction of Worldwide Energy Resources (POWER project) | https://power.larc.nasa.gov/data-access-viewer/ (accessed on 13 February 2023) |
| 2 | Soil data | FAO Digital Soil Map of the World (DSMW)-ESRI shapefile format | Food and Agriculture Organization of the United Nations | https://data.apps.fao.org/map/catalog/srv/eng/catalog.search#/home (accessed on 13 February 2023) |
| 3 | DEM data | Terra Advanced Spaceborne Thermal Emission and Reflection Radiometer-Global Digital Elevation Model (ASTER-GDEM)-Version 3-Grid format at 30m resolution | NASA's Earth Observing System Data and Information System (EOSDIS) | https://search.earthdata.nasa.gov/download/ (accessed on 13 February 2023) |
| 4 | LULC data | ESRI 2020 Global Land Use Land Cover from Sentinel-2 (TIF file format) | The map is derived from ESA Sentinel-2 imagery at 10 m resolution. | https://livingatlas.arcgis.com/landcover/ (accessed on 13 February 2023) |

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
