# Peer review of "Soil Water Erosion Modeling in Tunisia Using RUSLE and GIS Integrated Approaches and Geospatial Data"

_land, doi:10.3390/land12030548_

Round 1

Reviewer 1 Report

I have reviewed the manuscript titled " Modeling of soil water erosion in Tunisia using geospatial data and integrated approach of RUSLE and GIS". This manuscript discusses the soil water erosion in Tunisia using GIS.

The manuscript describes work that is a very important component to ecological work associated with the coastal area. Educating and optimizing the implementation of multi-purpose soil erosion or any other ecological systems is paramount of stakeholder acceptance and long-term success.

The abstract of the paper is mostly textual and introduces the approach used in research well enough. However, fails to provide insights on the findings or shortcomings of the findings or anything more that can be useful. I strongly believe that writing the abstract to reflect the paper in its entirety will be a good addition.

The keywords are plain, it will be interesting if the authors can come up with a set of more pertinent keywords that can allow a further insight into the work done here.  

1.           The MS does contribute scientifically or new in terms of methodology - a set of well-known methods are applied and used in this study to evaluate the soil erosion with GIS methods.

2.           I see a fruitful discussion on the generated datasets and scientific problem is analyzed and solved.

3.           The introduction is week and needs improvement, the method section is written well and Discussion is existing. Most of the literature cited is about soil erosion, however, there are more recent literature is required to discuss the Geospatial technique used in the past.

4.           I don't feel qualified to judge about the English language and style but the English language needs improvement.

Overall authors have applied useful statistics to extract useful results towards environmental significance.  

Recommendation.

Summarize my comments, I would recommend to Authors to address the revision and this manuscript can be published.

Author Response

We thank the reviewer for the constructive comments on our manuscript.

We sincerely appreciate the valuable feedback made by the reviewer that we took seriously to improve the last version of the manuscript.

Reviewer 2 Report

land-2225425-review

Modeling of soil water erosion in Tunisia using geospatial data and integrated approach of RUSLE and GIS

This article is well written and has high research value. Some suggestions are as follows:

1. The information shown in Figure 1 and Figure 2 overlap. It is recommended to delete Figure 1 and keep Figure 2.

2. Table 1 is recommended as an appendix rather than in the text.

4. The author needs to briefly explain the engineering or environmental implications of these findings, as well as existing problems and future research directions.

5. Two references may be helpful,

https://doi.org/10.1007/s10064-021-02123-7,

https://doi.org/10.1134/S1064229318120050

6. The conclusion part needs to be simplified, reflecting innovative results and future prospects.

7. Overall, this is a valuable topic to study. The authors also conducted detailed research and came to valuable conclusions that could be accepted after minor revision.

Author Response

Point 1: The information shown in Figure 1 and Figure 2 overlap. It is recommended to delete Figure 1 and keep Figure 2.
Response 1: We appreciate the reviewer's suggestion, but we believe that removing figure 1 would be unnecessary given that this figure presents in fact the geographical location of the study area (The whole Tunisian country) in relation with the African continent and the Mediterranean Sea, while figure 2 
presents the distribution of altitudes in Tunisia based on ASTER DEM data.

Point 2: Table 1 is recommended as an appendix rather than in the text.
Response 2: We have moved Table 1 to appendix A as requested by the reviewer with whom we agree since it can be considered as an additional data.

Point 3: The author needs to briefly explain the engineering or environmental implications of these findings, as well as existing problems and future research directions.
Response 3: We believe that environmental implications of the study results, as well as existing problems and future research directions, as the reviewer suggested, should be mentioned. Thus, we added the following paragraph in the conclusions (last paragrah): “The quantitative evaluation of spatial distribution of erosional area and erosion factor analysis are an important basis for decision 
making [75]. According to the study results, great efforts are needed to establish suitable strategies and measures to preserve water and soil resources and enhance ecological restoration in Tunisia. Among the research to be developed in the future, it is necessary to estimate other kinds of soil erosion such as 
wind erosion which is an important factor in the south part of Tunisia.”

Point 4: Two references may be helpful,
https://doi.org/10.1007/s10064-021-02123-7,
https://doi.org/10.1134/S1064229318120050
Response 4: We downloaded the two bibliographical references proposed by the reviewer but unfortunately we found that the first one “Yang, B., Liu, J. et al. Influence of fibers on desiccation cracks in sodic soil. Bull Eng Geol Environ 80, 3207–3216 (2021). https://doi.org/10.1007/s10064-021-02123-7” does not have a direct link with the subject of our article. On the other hand, the second one “Yuan, S., Ji, C., Li, X., Jia, Y. et al. Dynamic Assessment of Soil Water Erosion in the Three-North Shelter Forest Region of China from 1980 to 2015. Eurasian Soil Sc. 51, 1533–1546 (2018). https://doi.org/10.1134/S1064229318120050” is interesting and we have integrated it in the last paragraph of the conclusions.

Point 5: The conclusion part needs to be simplified, reflecting innovative results and future prospects.

Response 5: We have added in the last paragrah of the conclusions some future prospects, as the reviewer suggested: “Among the research to be developed in the future, it is necessary to estimate other kinds of soil erosion such as wind erosion which is an important factor in the south part of Tunisia.”  

Reviewer 3 Report

General comments:

The manuscript, entitled "Modeling of soil water erosion in Tunisia using geospatial data and integrated approach of RUSLE and GIS" evaluated regional soil erosion risk through the derivation of appropriate factors, using the Revised Universal Soil Loss Equation (RUSLE), which was applied to establish a soil erosion risk map of the whole Tunisian territory and to identify the vulnerable areas of the country. There are some issues with this manuscript, mainly related to the readability and composition of the manuscript.

 Please review the quality of your English throughout the manuscript.

Generally, I recommend to the authors in the introduction part to add more literature review in order to emphasize the need and originality of your research.

Specific comments:

Point 1: Line 369; I suggest that write separate results and discussion. 3. Results and 4. Discussion.

Point 2: Discussion section: This is an important part of the study, and you should write at least three to four pages discussing and comparing your findings, as well as arguing your findings with other similar research to emphasize the originality and novelty of the research.

Overall, the study is interesting, but a minor revision of the entire manuscript is required for publication in this journal. As a result, I recommend reconsideration after a minor revision of the manuscript.

Author Response

Point 1: Line 369; I suggest that write separate results and discussion. 3. Results and 4. Discussion.

Point 2: Discussion section: This is an important part of the study, and you should write at least three to four pages discussing and comparing your findings, as well as arguing your findings with other similar research to emphasize the originality and novelty of the research.

Responses 1 and 2: We appreciate the reviewer's suggestion, but we believe that the separation of results and discussion could lead us to modify the numbers and positions of several bibliographic references. Thus, to avoid making major changes of the manuscript, we prefer to keep the current structure, if the reviewer and the editor can allow it.
